

# Emissions databases for polycyclic aromatic compounds in the Canadian Athabasca Oil Sands Region – development using current knowledge and evaluation with passive sampling and air dispersion modelling data

Xin Qiu[1], Irene Cheng[2], Fuquan Yang[1], Erin Horb[3], Leiming Zhang[2], Tom Harner[2]

[1]Novus Environmental Inc., Guelph, Ontario, N1G 4T2, Canada
[2]Air Quality Research Division, Science and Technology Branch, Environment and Climate Change Canada, Toronto, Ontario, M3H 5T4, Canada
[3]Novus Environmental Inc., Calgary, Alberta, T2R 1K7, Canada

*Correspondence to*: Xin Qiu (xinq@novusenv.com) and Leiming Zhang (leiming.zhang@canada.ca)

**Abstract.** Two speciated and spatially-resolved emissions databases for polycyclic aromatic compounds (PAC) in the Athabasca oil sands region (AOSR) were developed. The first database was derived from volatile organic compound (VOC) emissions data provided by the Cumulative Environmental Management Association (CEMA) and the second database was derived from additional data collected within the Joint Canada-Alberta Oil Sands Monitoring (JOSM) program. CALPUFF modelling results for atmospheric polycyclic aromatic hydrocarbons (PAH), alkylated PAH, and dibenzothiophenes (DBT), obtained using each of the emissions databases, are presented and compared with measurements from a passive air monitoring network. The JOSM-derived emissions resulted in better model-measurement agreement in the total PAH concentrations and for most PAH species concentrations, compared to results using CEMA-derived emissions. At local sites near oil sands mines, the percent error of the model compared to observations decreased from 30% using the CEMA-derived emissions to 17% using the JOSM-derived emissions. The improvement at local sites was likely attributed to the inclusion of updated tailings pond emissions estimated from JOSM activities. In either the CEMA-derived or JOSM-derived emissions scenario, the model underestimated PAH concentrations by a factor of 3 at remote locations. Potential reasons for the disagreement include forest fire emissions, re-emissions of previously deposited PAHs, and long-range transport not considered in the model. Alkylated PAH and DBT concentrations were also significantly underestimated. The CALPUFF model is expected to predict higher concentrations because of the limited chemistry and deposition modelling. Thus the model underestimation of PACs is likely due to gaps in the emissions database for these compounds and uncertainties in the methodology for estimating the emissions. Future work is required that focuses on improving the PAC emission estimation and speciation methodologies and reducing the uncertainties in VOC emissions which are subsequently used in PAC emissions estimation.





# 1 Introduction

Polycyclic aromatic compounds (PACs) are a ubiquitous class of contaminants found naturally in geological deposits and produced as a byproduct of incomplete combustion of organic material (Baek et al., 1991; Boström et al., 2002; Neff et al., 2005). The broad PAC chemical classification includes hundreds of organic compounds that contain two or more fused benzene rings (Keyte et al., 2013; Kim et al., 2013). PACs include not only polycyclic aromatic hydrocarbons (PAHs), which have been the focus of previous scientific investigations (Timoney and Lee, 2011; Jautzy et al., 2013; Galarneau et al., 2014a,b; Hsu et al., 2015), but also alkylated PAHs, parent- and alkylated-dibenzothiophenes (DBTs) and other heterocyclic aromatic compounds containing nitrogen, sulphur or oxygen (Giesy et al., 2010; Schuster et al., 2015; Jariyasopit et al., 2016; Manzano et al., 2017). Exposure to some PACs has led to various carcinogenic, teratogenic and genotoxic effects in animals and humans and cases of skin and eye irritation and inflammation (ATSDR, 2009; CCME, 2010; Kim et al., 2013; Wickliffe et al., 2014; Zhang et al., 2015). Detailed toxicity information on individual PAC species have not been elucidated because subjects have mainly been exposed to a mixture of compounds (Gosselin et al., 2010; Kim et al., 2013; Jariyasopit et al., 2016). Limited toxicology data suggests some alkylated PAHs and heterocyclic compounds are more deleterious than the unsubstituted compounds (Yu, 2002; Rhodes et al., 2005; Turcotte et al., 2011; Lin et al., 2015). Alkylated PACs are not as widely studied as unsubstituted PACs; however, given the equivalent or increased potential for toxic effects, further study is warranted. It has also been observed that alkylated PAHs and DBTs are more abundant in petrogenic sources (Rhodes et al., 2005; Yang et al., 2011; Wickliffe et al., 2014) making it important to study PACs in oil sands regions.

Situated in Canada's boreal forest, the Athabasca oil sands region (AOSR) of northern Alberta is a concentrated area of industrial development with numerous facilities extracting and processing bitumen. This region makes up ~82% of the total bitumen in the oil sands deposits of northeastern Alberta, of which 20% (4,800 km$^2$) can be extracted by surface mining (Small et al., 2015). PAC emissions sources directly related to oil sands development include bitumen production facilities, mine face, mine fleet, and tailings ponds (Parajulee and Wania, 2014). Sources from non-industrial activities also contribute: wood burning, forest fires, and vehicular emissions have also been identified as sources of pervasive airborne PAHs (Hsu et al., 2015).

PAC emissions inventories for the AOSR are necessary for the modelling of PAC concentrations, deposition and subsequent assessments of ecosystem impacts. The Canadian National Pollutant Release Inventory (NPRI) contains speciated PAH emissions from point and fugitive sources in the AOSR; however, only the annual total facility emissions are required to be reported. There are two other emissions databases in the AOSR that are suitable for compiling a PAC emissions database. The Cumulative Environmental Management Association (CEMA) and Joint Canada-Alberta Oil Sands Monitoring (JOSM) emissions databases include spatially-resolved volatile organic compound (VOC) emissions data from additional source categories (e.g., tailings ponds, mine face, mine fleet, non-industrial), but not for PAC species. Thus, a comprehensive PAC emissions database needs to be developed that can provide speciated as well as spatially-resolved emissions data suitable for air quality modelling.



Quantifying the PAC emissions from the AOSR remains a significant challenge because of uncertainties in the emissions from oil sands production. A study estimated that PAH fluxes from tailings ponds were 4.6 times greater than the point source and fugitive emissions reported by the oil sands industry to the NPRI in 2012 (Galarneau et al., 2014a). Model simulations considering only direct air emissions underestimated phenanthrene, pyrene and benzo(*a*)pyrene concentrations in air, water, soil and foliage, whereas simulations including both direct air emissions and tailings pond emissions were more comparable to observations (Parajulee and Wania, 2014). Another source of airborne PAHs that has not been included in the emissions inventory is petroleum coke stockpiles in the mining areas, which can be resuspended by wind and deposited (Zhang et al., 2016). Analysis of wildlife samples near oil sands development indicate moose and wolves have been exposed to alkylated PAHs from petrogenic sources (Lundin et al., 2015). This study suggested that PACs are making their way through ecosystems in northern Alberta. However, the uncertainties in PAC emissions in this region need to be resolved in order to improve the understanding of how the emissions are impacting ecosystems.

In this study, two PAC emissions databases were developed. In the CEMA-derived emissions database, PAH emissions were estimated from PAH speciation profiles and CEMA emissions data which included VOC emissions from oil sands mining areas and non-industrial sources. In the JOSM-derived emissions database, several sources of data obtained from the JOSM program were used to estimate PAC emissions (i.e. PAH, alkylated PAH and DBT) including: VOC emissions from oil sands mining areas and non-industrial sources from the JOSM emissions database, tailings ponds emissions estimates (Galarneau et al., 2014a), and passive air concentrations (Schuster et al., 2015). The CALPUFF atmospheric dispersion model was then used to simulate PAC concentrations in the AOSR. Model simulations were conducted using the emissions from the CEMA-derived database in one scenario and the JOSM-derived database in another. The modeled concentrations of PAHs, alkylated PAHs and DBTs were compared with passive monitoring data to assess which emissions input can achieve better model-measurement agreement.

## 2 Methodology

### 2.1 Development of PAC emissions databases

#### 2.1.1 CEMA and JOSM emissions databases

CEMA comprises aboriginal, government, non-governmental organizations, and industry stakeholders. CEMA's role includes developing air quality management frameworks/plans for the Regional Municipality of Wood Buffalo (RMWB). The implementation of these frameworks/plans is supported by ambient air quality and deposition modelling, which assesses the current and future environmental impacts of emissions from oil sands development and other local or regional sources in the RMWB including the AOSR (Vijayaraghavan et al., 2010). The models require a representative regional emissions database. The focus of the CEMA emissions database was to identify and quantify industrial and non-industrial emission sources in the AOSR. Industrial sources are comprised of stacks, mine





fleet exhausts, fugitive plant, fugitive mine pit, fugitive tailings management, while non-industrial sources include community, highway (on-road), and recreational vehicle (off-road) sources (CEMA, 2011). This database is based on emissions data from 2009 to 2010.

5 The JOSM emissions database (ECCC and AEP, 2016) was developed by the Governments of Canada and Alberta. This database covers the oil sands areas and is based partially on existing emissions data from NPRI and CEMA. The JOSM database used in this study is based on the data available up to October 31, 2014 (ECCC, 2016). Neither the CEMA nor JOSM emissions database contain individual PAC species or total PACs. Instead, the databases report total VOCs, which includes PAHs and other hydrocarbons. Speciated PAH air emissions have been reported

10 by the oil sands industries to the NPRI for point and fugitive sources (ECCC, 2017); however the actual source locations of the boilers, heaters, co-generation units, etc. belonging to each facility and stack dimensions and flow parameters are not required to be reported. Because these physical specifications are necessary to accurately model air pollution dispersion, the PAH emissions from the NPRI are not suitable for this study. Recently, PAHs disposed in tailings and waste rock were reported to the NPRI; however, the fluxes to air are unknown.

In this study, PAC emissions to air were estimated for a broad range of source categories using the VOC emissions in Table S1 of the Supplement. In the JOSM database, VOC emissions from tailings ponds, mine face and point sources have been scaled up from the CEMA database using the 2010 NPRI data. As shown in Table S1, VOC emissions from mine fleet, residential, commercial, non-industrial, and line sources (transportation) were relatively

20 unchanged because the JOSM database adopted these emissions from the CEMA database. Oil sands mining emissions from the JOSM database are essentially the same as the CEMA database with partial updates for a few facilities using NPRI data and mining site spatial surrogates from satellite data. We assumed no changes to the point source VOC emissions because model sensitivity analysis indicated that point and line sources within the model domain have minimal impact on PAC concentrations in the oil sands mining areas (Sect. 3.2). The major point

25 sources are located south of the study area in the Southern Athabasca Oil Sands area, which is dominated by in-situ bitumen extraction. Geospatial data from the databases indicate that oil sands mining areas have increased from the CEMA to the JOSM database. The surface area of tailings ponds grew from 104.7 $km^2$ in the CEMA database to 182.6 $km^2$ in the JOSM database (Fig. S1 of the Supplement), while mine face areas increased from 35.1 $km^2$ to 170.1 $km^2$.

30 **2.1.2 Estimation of PAC emissions and speciation methodology**

CEMA-derived and JOSM-derived emissions for PACs were estimated for the following source categories: 1) tailings ponds; 2) mine face; 3) mine fleet; and 4) other sources including point sources, transportation, residential, and commercial. Different approaches were taken to estimate the PAH emissions from the various source categories; the details are described in Sect. S1 of the Supplement. For 1) tailings ponds, the PAH emission

35 speciation was based on the study by Galarneau et al. (2014a). This study reported emissions of 1,069 kg $yr^{-1}$ from tailings ponds for 13 PAH species during the JOSM field campaign (2010-2012). The annual emissions were



distributed between the individual tailings ponds using the area of the tailings ponds. For 2) mine face PAH emissions, an assumption was made that the emission flux of PAH species volatilized from the mine face would be lower than that of tailings ponds based on the ratio of the VOC emissions from these sources. There were no direct emissions measurements available from a mine face in the oil sands area at the time of this study. For 3) mine fleet,

PAHs were speciated by mass fraction of total VOC emissions from mine fleet based on the CEMA study (Vijayaraghavan et al., 2010). CEMA's mine fleet PAH speciation profiles were developed using the USEPA SPECIATE database (USEPA, 2017). For 4) other emissions, speciation of both point and non-point source emissions was based on VOC and $PM_{2.5}$ emissions and speciation profiles in the CEMA study by Vijayaraghavan et al. (2010). Note that the majority of CEMA speciation profiles were based on a series of environmental impact

assessment studies in the oil sands area (Vijayaraghavan et al., 2010). For those point and non-point sources that were not available in the CEMA database, PAH species were estimated using SPECIATE (USEPA, 2017), which has a repository of organic and PM speciation profiles for various air pollution sources (Simon et al., 2010). The profiles can be used to create speciated emissions inventories for ozone modelling (e.g. NO, $NO_2$, and explicit VOC species) and estimate hazardous and toxic air pollutant emissions from total PM and organic primary emissions. For

alkylated PAHs and DBTs, emissions from mine fleet and transportation were estimated using SPECIATE, while an approach using the ratio of total PAH to alkylated PAH or DBT from passive sampling data was used to calculate tailings pond emissions. Based on the expansion of the tailings ponds and mine face surface areas from the CEMA to JOSM databases which in turn led to higher VOC emissions from these sources (Table S1 and sect. S1 of the Supplement), most of the PAH emission increases are attributed to tailings ponds and mine face sources. The

estimated emissions of total PAHs, alkylated PAHs, and DBTs from the various source categories are shown in Table 1.

It was found that there were still gaps in the existing emissions database, and speciation profiles were largely missing particularly for alkylated PAHs and DBTs. In this paper, the focus is on PAHs; however, alkylated PAHs and DBTs were still modelled despite the limited knowledge of emissions profiles.

**2.2 CALPUFF model**

The USEPA CALPUFF model was run using the CEMA-derived or JOSM-derived emissions database for PACs, and the PAC concentrations downwind were predicted. CALPUFF is an advanced, integrated Lagrangian puff modelling system for the simulation of atmospheric pollution dispersion adopted by the USEPA in its Guideline on Air Quality Models and accepted by Alberta Environment and Parks and the Alberta Energy Regulator. CALPUFF

takes three-dimensionally varying wind, temperature and turbulence fields from the CALMET model which is a stand-alone meteorological data processor. In this study, CALMET harmonizes Weather Research and Forecasting (WRF) generated three-dimensional data and local observed data together. The WRF Model is a next-generation mesoscale numerical weather prediction system designed for both atmospheric research and operational forecasting needs (Skamarock et al., 2008).



CALPUFF was set up and modelled following the Alberta Air Quality Model Guideline (AEP, 2013). The model was run from October 2010 to the end of 2012. For this study, the CALPUFF model simulated the dispersion and transport of PACs and estimated the ambient air concentrations. However, the dry and wet deposition schemes were not activated in the model, and the chemical processes were limited for modelled PAC species. The model predicted

the concentrations of 16 USEPA priority PAHs: naphthalene (NAPH), acenaphthylene (ACY), acenaphthene (ACE), fluorene (FLR), phenanthrene (PHEN), anthracene (ANTH), fluoranthene (FLRT), pyrene (PYR), benz[a]anthracene (BaA), coeluting chrysene and triphenylene (CHRYþTRIP), benzo[b]fluoranthene (BbF), benzo[k]fluoranthene (BkF), benzo[a]pyrene (BaP), indeno[1,2,3-cd]pyrene (I123cdP), dibenz[a,h]anthracene (dBahA) and benzo[ghi]perylene (BghiP), as well as total alkylated PAHs and DBTs. Note that retene (methyl

isopropyl phenanthrene) was categorized as part of total alkylated PAHs.

CALPUFF modelling was applied to both discrete receptors (i.e. sensitive receptors) and gridded receptors (i.e. CALPUFF modelling grids). The CALPUFF modelling domain covers a large area bounded by the following coordinates, SW: 54.599, -114.000, SE: 54.595, -107.807, NW: 59.766, -114.450, NE: 59.760, -107.328. The model

domain is larger than the study area, which is focused on the oil sands mining areas (Fig. S2). It covers all possible sources including traffic and transportation along the road network, industrial areas, residential/commercial sources, and the northern part of Edmonton (urban area). Emissions outside the model domain are not accounted for in the model. Further details regarding the CALPUFF model settings and options are provided in Sect. S2 of the Supplement.

**2.3 Model evaluation against passive monitoring data**

Model-predicted PAC concentrations were compared with measurements from a 17-site passive air sampling network (Fig. S1; Harner et al., 2013; Schuster et al., 2015). The model evaluation domain focused on a specific area (Fig. S1a; SW: 56.272, -112.260, SE: 56.278, -110.452, NW: 57.880, -112.315, NE: 57.885,-110.428). Figure S1b illustrates the locations of the passive air samplers in the AOSR. The PAC data was collected from 172 samples

at 17 sites between November 2010 and June 2012. There are 8 local sites (L) which are accessible by road and near oil sands operations and 9 remote sites (R). PAH concentrations were relatively constant throughout most of the sampling period, except for elevated concentrations observed from April to July 2011 which were attributed to forest fires events (Schuster et al., 2015). The forest fire events were identified based on high retene concentrations and retene/(retene+chrysene) ratio approaching one (Schuster et al., 2015). Biomass, fossil fuel, and petrogenic

combustion can also be distinguished based on fluoranthene/(fluoranthene+pyrene) ratio (Lundin et al., 2015). Data collected during the forest fire period were excluded from the modelling evaluation because PAH emissions from forest fires were not inputted into the model. Additionally, site L14 showed extremely high PAH concentrations during the summer months, which was also excluded from model evaluation. The higher summertime concentrations at site L14 was likely due to revolatilization of PAHs from nearby Gregoire Lake (Hsu et al., 2015).

Furthermore, due to the high volatility of NAPH leading to sampling biases (Harner et al., 2013) and the high NAPH



concentrations (one to two orders of magnitude higher than all other PACs), we took NAPH out from the total PAH group and treated it separately to avoid masking the other PAC species.

## 3 Results and Discussion

### 3.1 Total PAH concentrations

In this study, total PAHs included all PAHs measured during the JOSM program in 2010-2012 except NAPH as mentioned above. Figure 1 shows a comparison of the average total PAH concentrations between the CALPUFF model and passive measurements at local and remote sites. Site L14 and data that had been impacted by forest fires were excluded in Figure 1 as explained above. The modelled results included two emissions input scenarios: CEMA-derived and JOSM-derived PAC emissions. Note that the CALPUFF air quality modelling runs used the

same meteorological data input from CALMET and the same CALPUFF model settings. The only difference was the use of different PAC emissions data.

Overall, it can be seen that the model performed much better at local sites than at remote sites as shown in Figures 1a and 1b, respectively. CALPUFF was capable of reproducing the passive measurements at the local sites

particularly at L04, L06 and L13, but underestimated PAHs considerably at remote sites except at R05. Figure 1 also suggests that model-JOSM case performed better than the model-CEMA case at most of the sites, except R05. These results suggest that the improvements to the JOSM-derived emissions database led to better agreement between model and observations than the CEMA-derived emissions database.

In terms of the model performance, the model percentage errors at local sites were much smaller than remote sites: 17% vs. 66% with JOSM-derived emissions, and 30% vs. 67% with CEMA-derived emissions (Table S2). While model-JOSM performed better than model-CEMA at the local sites, little improvement was found at the remote sites. Modelled concentrations produced using either the CEMA-derived or JOSM-derived emissions data were underestimated by a factor of 3. This suggests that the changes in oil sands emissions from CEMA-derived to

JOSM-derived database had essentially an insignificant impact on modelling results in the remote area. Model underestimation of PAH concentrations at most of the remote sites could be due in part to small forest fires in the remote area. Based on the fire radiative power (FRP) data from MODIS (NASA, 2017), 14 of the 17 passive sampling sites were strongly impacted by forest fires from April to July 2011 (Fig. S3a). Thus, the passive measurements collected during this period were omitted from the model evaluation. During other times of the year

in 2011 and 2012, most of the sites were unaffected by large forest fires although the R01, R08 and R09 remote sites may have been affected by small fires nearby (Fig. S3b). Besides forest fires, elevated regional background levels of PAHs in air from long-range transport of emissions and re-emissions to air of previously deposited PAHs that are not accounted for in the model could explain the underestimated concentrations at remote sites. While the model did not include long-range emissions transport, the lack of deposition loss of PAHs in the model may partially





compensate for the missing background emissions in the model. If deposition had been considered in the model, the modeled concentrations would be even lower than the current predictions.

### 3.2 Spatial distribution of PAH concentrations

Figure 2 illustrates the spatial distribution of the model-predicted average PAH concentrations using CEMA-derived

(2a) and JOSM-derived (2b) emissions, overlaid with passive measurements from November 2010 to June 2012. The contours in Figure 2 were produced from the model outputs; the coloured dots represent PAH measurements at the 17 passive sampling sites; the colour legend and the scale represent PAH concentrations in ng m$^{-3}$. If the contour colour matches the dot colour, the model is able to reproduce the measured data.

A comparison of Figures 2a and 2b shows that the model-JOSM (2b) reproduced the elevated PAH concentrations over major mining areas, such as areas south of Fort McKay, Mildred Lake settling basin, tailings ponds owned by Suncor Energy, and tailings ponds located north of Fort McKay owned by Syncrude Canada Ltd. Model-JOSM was the better model at most of the local sites (contour colours closely match the dot colours). Point sources and transportation emissions had minor impacts on modelled PAH concentrations according to the model sensitivity

analysis (Fig. S4). Although PAH emissions from point sources were greater than the emissions from other source categories (Table 1), this did not result in higher ground level concentrations near the point sources. For point sources, other factors such as stack heights, exit temperatures and exit velocities are also important to plume rise and dispersion, which can lead to lower ground level concentrations compared to those impacted by similar emissions from area sources such as tailings ponds. The impact at ground level from point sources is based on a combination

of factors, not only on the emission rates.

CALPUFF significantly underestimated the PAH concentrations in remote areas regardless of the emissions data input. High PAH concentrations at remote sites are unlikely to be subject to industrial emissions. Thus, there are likely other sources of PAHs, such as small forest fires that contributed to the elevated PAH concentrations and re-

25 volatilization of previously deposited PAHs, which were not considered in the model. The underestimation by the model could also be due to the underestimation of VOC emissions in the AOSR (Li et al., 2017), since PAC emissions from mine fleet, point sources, transportation, residential and commercial sources were derived from VOC emissions and speciation profiles.

The model-JOSM (Fig. 2b) output also suggests that high PAH concentrations were not necessarily located at local sites. For example, sites L01, L04 and L05 are located east of Syncrude Canada's Mildred Lake tailings facilities, while the area with highest PAH concentrations were found northeast of the monitoring sites. A similar effect was also observed at local sites west of the Suncor Energy tailings ponds.



### 3.3 PAH speciation analysis

Figures 3a and b illustrate the ratios of speciated PAH modelled-to-measured concentrations for all of the valid data pairs available for the model evaluations, with both CEMA-derived and JOSM-derived emissions. The ratios for local and remote sites are shown in Fig. 3a and b, respectively. CEMA and JOSM modelled concentrations and

measured concentrations averaged from all sites are illustrated in Fig. 3c and d.

For local sites, there were 5 PAH species (ANTH, BaA, CHRYþTRIP, BbF, and BghiP) from model-JOSM (orange dots) for which the model agreed with measured concentrations within a factor of 2. The average modelled-to-measured concentration ratios for these 5 species were 1.6, 0.9, 0.5, 0.5 and 1.1, respectively. Only 3 PAH species

(ANTH, BaA and BghiP) from model-CEMA (blue dots) yielded average modelled-to-measured concentration ratios that were close to the ideal value of unity (1.7, 0.7 and 1.5, respectively). Similar patterns were found at remote sites. Comparison of the modelled speciated PAH concentrations from all the sites (Fig. 3c,d) between the CEMA-derived and JOSM-derived emissions scenarios show there were minor differences in most of the PAH species, including NAPH, ACY, ACE, FLR, ANTH, FLRT, BkF, I123cdP, and dBahA. The model output using the

JOSM-derived emissions predicted higher concentrations that were closer to the observed concentrations than of the CEMA-derived emissions for PHEN, BaA, CHRYþTRIP, BbF, and BaP. For the species ACY, FLRT and PYR, both emissions scenarios overestimated the observed concentrations. The dominant PAH species from passive measurements were NAPH, PHEN, and FLR. In another study using high-volume sampling methods to measure PAHs from the AOSR, PHEN, NAPH, and ANTH were the most abundant species (Wnorowski, 2017). However in

our model simulations, the dominant PAH species were NAPH, PYR, and PHEN for the JOSM-derived emissions scenario and NAPH, PYR, and ACY for the CEMA-derived emissions scenario. The discrepancies in the dominant PAH species between the model and measurements suggest uncertainties in the PAH speciation profiles for oil sands sources.

### 3.4 Total alkylated PAH and DBT concentrations

The modelled concentrations of total alkylated PAH and DBT from the JOSM-derived emissions database were underpredicted compared to the measurements from the passive sampling network (Fig. 4). This could be due to (1) a lack of emissions estimates from other oil sands sources, such as mine face and facility fugitive emissions (e.g. Zhang et al., 2016), since alkylated PAH and DBT emissions were only estimated from mine fleet, line sources, and tailings ponds; and (2) uncertainties with using monitored concentration ratios ($R$) between PAHs and alkylated

PAHs/DBTs to back-calculate alkylated PAH and DBT emissions (sect. 1.5 in the SI). In this study, we assumed a constant average ratio for PAH/alkylated PAH and PAH/DBT; however, this ratio could change depending on where the monitoring sites are located because of other emission sources and the decline in PAC deposition with distance from major oil sands development areas (Manzano et al., 2016).



## 4 Conclusions

The JOSM-derived emissions database improved CALPUFF model predictions of total PAH concentrations against passive monitoring data at local sites compared to using the CEMA-derived emissions database. The model significantly underestimated PAH concentrations at most of the remote locations. Although the data impacted by

5 major forest fire events were excluded from model evaluation, it is possible that unreported small forest fires, re-emissions of previously deposited PAHs, and long-range transport contributed to the elevated PAH concentrations at remote sites. For alkylated PAHs and DBTs, the model underestimated the concentrations at all of the sites.

One of the emissions gaps identified in this study is a lack of emissions data on alkylated PAHs and DBTs.

Uncertainties in the methodology for estimating PAC emissions and speciation profiles of PACs from different oil sands emission sources are potential reasons for the discrepancies between model results and observations. These issues need to be resolved to better model the PAC concentrations and deposition in this region. Using a dispersion model such as CALPUFF with detailed 3D meteorological fields generated by WRF/CALMET to drive air dispersion from oil sands emissions sources can provide a better understanding of PAC spatial distribution patterns.

Model results can identify potential "hot spots" with the highest concentrations, which can be used to guide monitoring network design. For instance, modelling results from this study suggest the current PAH monitoring sites are not located within the highest modelled concentration areas which are adjacent to major tailings ponds and mines. The addition of an air–surface exchange parameterization should be evaluated as a potential response to the seasonally varying prediction capabilities of the model for the most volatile compounds.

**Data Availability**

The data from this study are accessible from the links provided in the references.

**Competing Interests**

The authors declare that they have no conflict of interest.

**Acknowledgements**

We acknowledge funding support from the Joint Canada-Alberta Implementation Plan for Oil Sands Monitoring (JOSM) program, Alberta Environment and Parks for providing some of the oil sands data used in the study, and the use of data and imagery from LANCE FIRMS operated by the NASA/GSFC/Earth Science Data and Information System (ESDIS) with funding provided by NASA/HQ. The authors thank Stewart Cober, Elisabeth Galarneau,
Jasmin Schuster, and Andrzej Wnorowski from ECCC for comments that helped improve the paper and Junhua Zhang from ECCC for advice on the JOSM emissions database.





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



**Table 1. Estimated PAH, alkylated PAH, and DBT emissions (kg yr$^{-1}$) over the model domain**

|  | PAH | | Alkylated PAH | DBT |
|---|---|---|---|---|
|  | CEMA-derived emissions | JOSM-derived emissions | JOSM-derived emissions | JOSM-derived emissions |
| Tailings pond | 417 | 1,069 | 2,442 | 255 |
| Mine face | 24 | 600 | na | na |
| Mine fleet | 9,573 | 9,698 | 7,596 | 0 |
| Residential and commercial | 58 | 58 | na | na |
| Non-industry (local traffic and airport) | 1,628 | 1628 | na | na |
| Point sources | 43,299 | 43,299 | na | na |
| Line sources | 1,401 | 1,401 | 7,200 | 3 |

na: not available

25





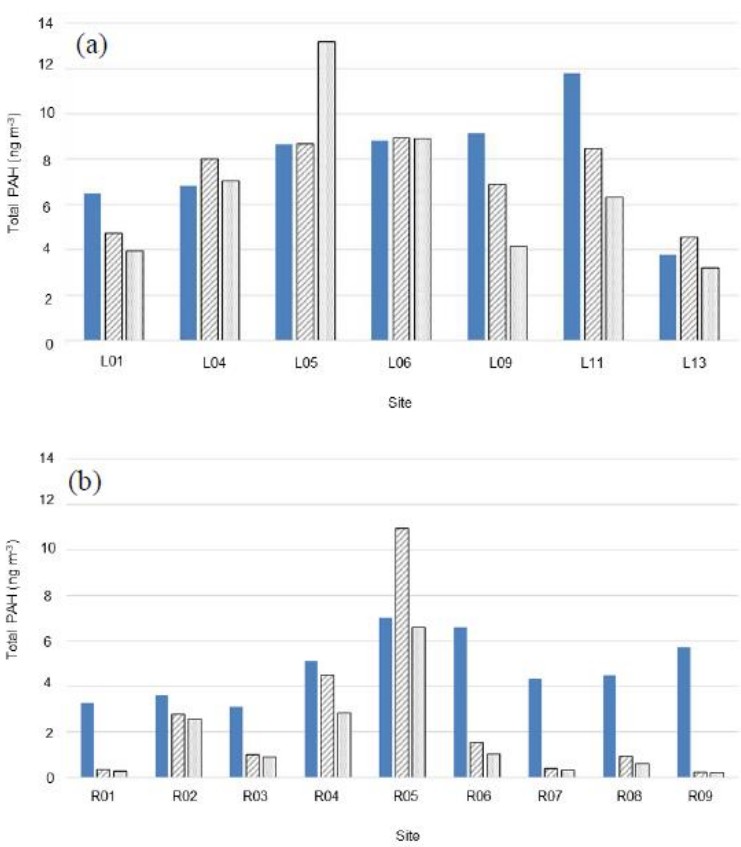

**Figure 1. Average total PAH concentrations (excluding NAPH) from November 2010 to June 2012 at local (a) and remote sites (b): passive measurements (blue), modelled-JOSM case (striped) and modelled-CEMA case (grey).**





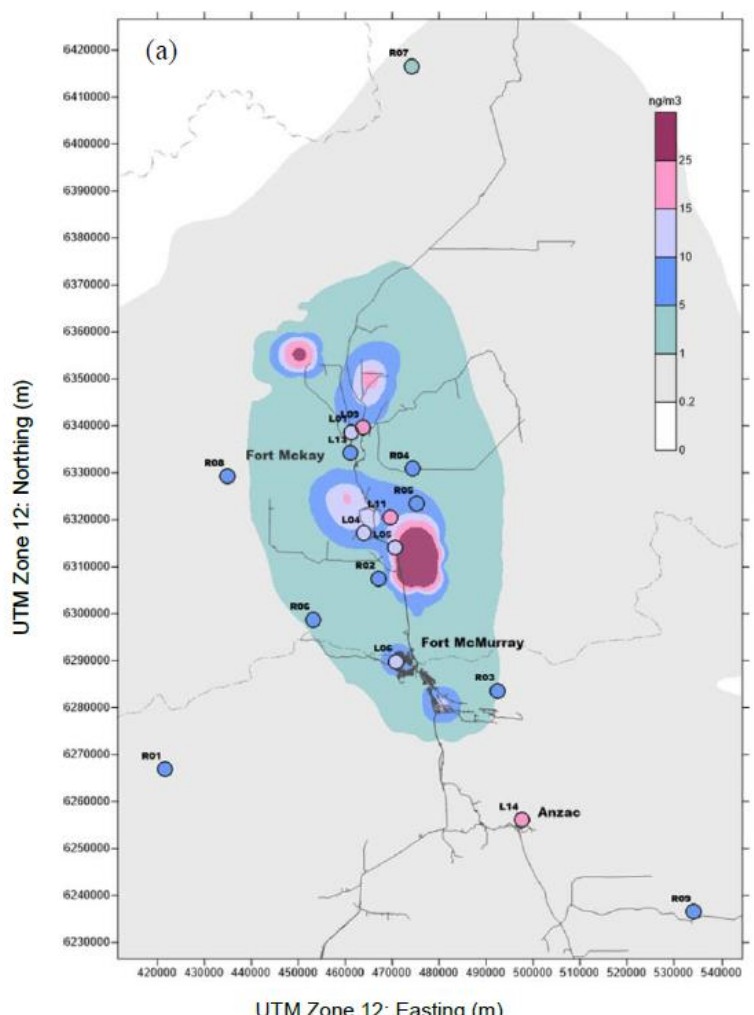

**Figure 2(a). Average model-predicted PAH concentration contours using CEMA-derived emissions overlaid with passive measurements from November 2010 to June 2012 (circles)**



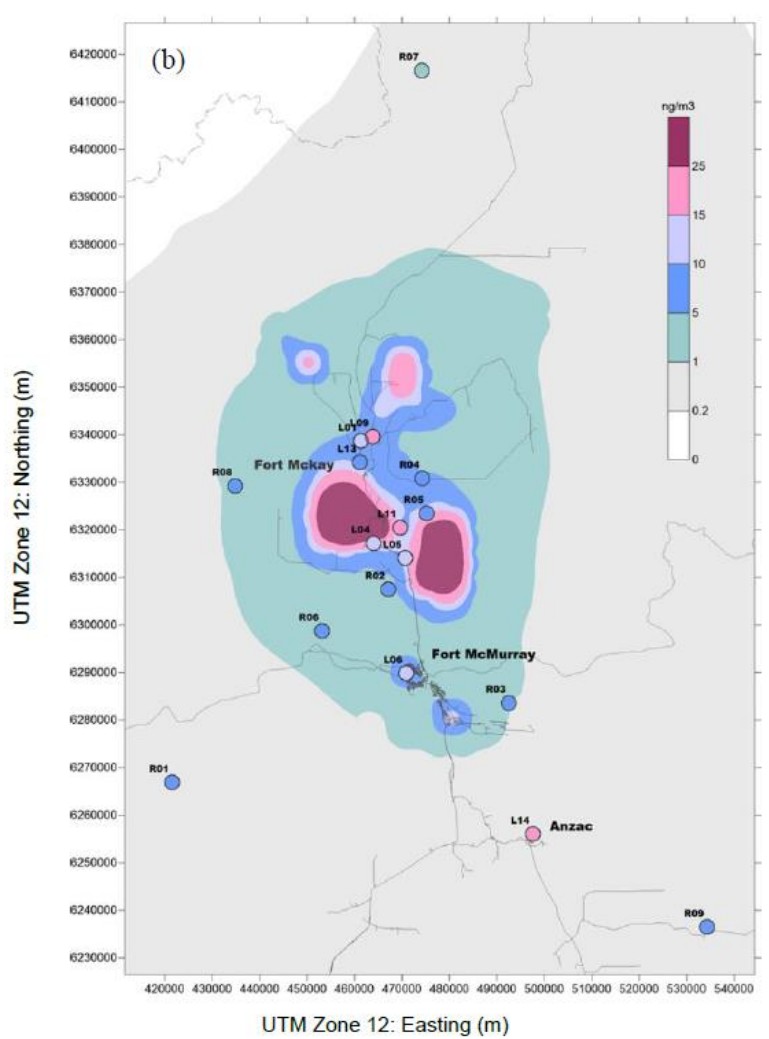

**Figure 2(b). Average model-predicted PAH concentration contours using JOSM-derived emissions overlaid with passive measurements from November 2010 to June 2012 (circles).**



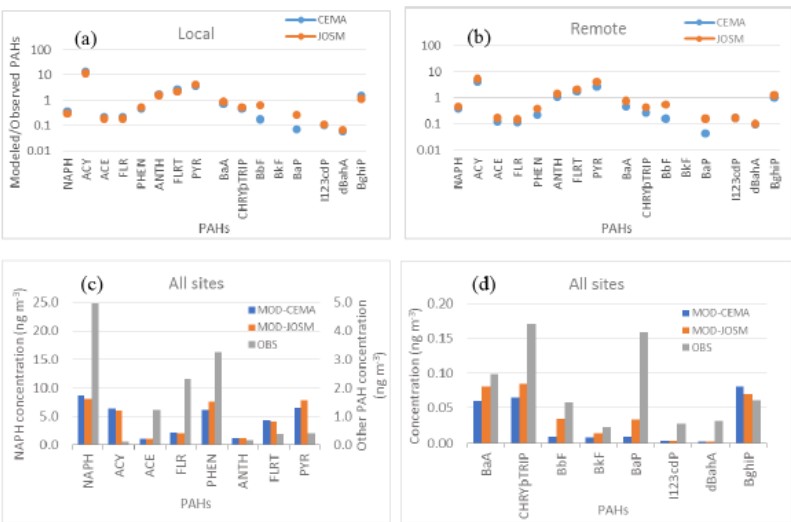

**Figure 3. The ratios of CEMA and JOSM modelled concentrations to observed concentrations of speciated PAHs at local (a) and remote (b) sites. Comparison of average CEMA and JOSM modelled concentrations and observed concentrations at all sites (c,d). Note the different y-axis scales for the concentrations in figures c and d.**



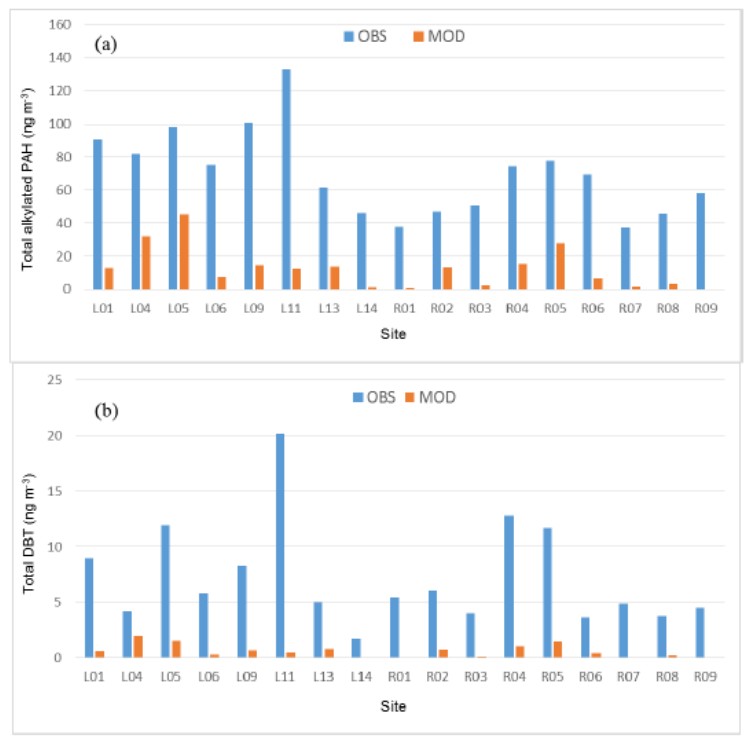

**Figure 4. Comparison of average concentrations of total alkylated PAHs (a) and DBTs (b) between passive sampling observations and model using JOSM-derived emissions database.**

