# Peer review of "Emissions databases for polycyclic aromatic compounds in the Canadian Athabasca Oil Sands Region – development using current knowledge and evaluation with passive sampling and air dispersion modelling data"

_Atmospheric Chemistry and Physics, 2017_

## Referee Comment (RC1) · Anonymous Referee #1 · 12 Jan 2018

This paper developed two speciated and spatially-resolved emissions databases for polycyclic aromatic compounds (PAC) in the Athabasca oil sands region (AOSR), and compared the two emissions databases with the measurements from a passive air monitoring network. Papers have a high degree of novelty and I recommend to publish after a minor revised. 1. Please delete some basic concepts, concise articles. 2. What are the PAHs in CEMA database and JOSM database, respectively? Are they the same? 3. In Results and Discussion, please describe the same meteorological data in

detail.
* * *

---

## Referee Comment (RC2) · Anonymous Referee #2 · 17 Jan 2018

Within the article, two speciated and spatially-resolved emissions databases for PACs in AOSR were developed. Further, the PAC concentrations in AOSR were simulated using the CALPUFF atmospheric dispersion model for both scenarios (both databases) and compared with passive monitoring data to assess which emissions input can achieve better agreement with measurements. According to my opinion, the manuscript represent a significant scientific contribution in studying PACs (PAHs, alkylated PAHS and DBTs) in oil sends regions where uncertainties in the PACs emissions

are still significant. I recommend the manuscript for publication with minor revision: 1. Although if deposition had been considered in the CALPUFF model, the modeled values would be even lower then the measured, I would ask the authors to explain why they have excluded the loss by wet and dry deposition in the modeling process. Were there any other reasons? 2. The authors should also take into consideration the fact that values of PAC concentrations obtained using the passive samplers refer only to the gaseous phase of pollutant and reflect a more accurate concentration for the low molecular weight PACs compering to high molecular weight compounds.

---

## Referee Comment (RC3) · Anonymous Referee #3 · 17 Jan 2018

This research work compared CALPUFF modelling results applying the two air emissions databases of CEMA and JOSM programs. The modelling results are then compared with observations to evaluate accuracy of the air emissions values. This research makes significant contribution to the work of PAHs air emission estimation in the oil sands region. While dispersion models could have systematic error existing inherently in the model, particularly and usually lead to underestimation at low pollutant concentrations, this research presents a progressive approach to compare the mod-

elling results relatively for the original emissions data and the improved one. I suggest to publish it to make colleagues working in this field be aware of the work progress.

It would be clearer if the author could add more information on meteorology and emission summary. Additionally, PAHs have a wide spectrum including compounds in gaseous phase and particulate phase, which can exhibit different characteristics during transport and deposition. Although the research is focused on relative comparison of two emissions databases with only considering dispersion, the author may analyze qualitatively the resultant impact of turning off deposition modelling on modelling results in general.

---

## Author Comment (AC1) · 23 Jan 2018

Response to Reviewer #1

We appreciate the comments by the reviewer to help us improve the paper. Our responses to the specific comments are shown below.

This paper developed two speciated and spatially-resolved emissions databases for

[Figure]

polycyclic aromatic compounds (PAC) in the Athabasca oil sands region (AOSR), and compared the two emissions databases with the measurements from a passive air monitoring network. Papers have a high degree of novelty and I recommend to publish after a minor revised.

1. Please delete some basic concepts, concise articles.

Response: We condensed and deleted some of the basic information (e.g. basics about PAHs in the first paragraph of the introduction) in the revised paper to keep it concise.

2. What are the PAHs in CEMA database and JOSM database, respectively? Are they the same?

Response: The 16 parent PAHs are the same in the CEMA-derived and JOSM-derived emissions databases. As mentioned in sect. 2.2, the PAHs include: naphthalene (NAPH), acenaphthylene (ACY), acenaphthene (ACE), fluorene (FLR), phenanthrene (PHEN), anthracene (ANTH), fluoranthene (FLRT), pyrene (PYR), benz[a]anthracene (BaA), coeluting chrysene and triphenylene (CHRY_TRIP), benzo[b]fluoranthene (BbF), benzo[k]fluoranthene (BkF), benzo[a]pyrene (BaP), indeno[1,2,3-cd]pyrene (I123cdP), dibenz[a,h]anthracene (dBahA) and benzo[ghi]perylene (BghiP). However, total alkylated PAHs and dibenzothiophenes (DBT) emissions were only estimated for the JOSM-derived emissions database, since the monitoring of these additional compounds is part of monitoring activities under the JOSM program. This will be clarified in the revised paper.

3. In Results and Discussion, please describe the same meteorological data in detail.

Response: The meteorological data input for the CALMET model was described in sect. 2.2. A more detailed description will be provided in the revised paper.

CALPUFF takes three-dimensionally varying wind, temperature and turbulence fields from the CALMET model. The 3-D winds and temperature fields from CALMET are

reconstructed using meteorological measurements, orography and land use data. Besides wind and temperature fields, CALMET determines the 2-D fields of micrometeorological variables needed to carry out dispersion simulations (mixing height, Monin Obukhov length, friction velocity, convective velocity, etc.). A two-step approach is typically used to compute the wind fields in CALMET. In the first step, an initial guess wind field is adjusted for kinematic effects of terrain, slope flows, and terrain blocking effects to produce a Step 1 wind field. The second step applies an objective analysis procedure to introduce observational data into the Step 1 wind fields to produce the final wind fields. In this study, CALMET used the Weather Research and Forecasting (WRF) model due to its capability of simulating regional flows and certain aspects of local meteorological conditions such as complex terrain. It replaces the two-step approach because of the higher spatial resolution of the WRF output compared to observational data. The output of the CALMET model is directly interfaced with the CALPUFF dispersion model for further air quality modelling.
* * *

---

## Author Comment (AC3) · 23 Jan 2018

Response to Referee #3

We appreciate the comments by the reviewer to help us improve the paper. Our responses to the specific comments are shown below.

This research work compared CALPUFF modelling results applying the two air emissions databases of CEMA and JOSM programs. The modelling results are then compared with observations to evaluate accuracy of the air emissions values. This research makes significant contribution to the work of PAHs air emission estimation in the oil sands region. While dispersion models could have systematic error existing inherently in the model, particularly and usually lead to underestimation at low pollutant concentrations, this research presents a progressive approach to compare the modelling results relatively for the original emissions data and the improved one. I suggest to publish it to make colleagues working in this field be aware of the work progress.

It would be clearer if the author could add more information on meteorology and emission summary.

Response: A summary of the meteorological model that drives the CALPUFF model will be provided in sect. 2.2 of the revised paper.

CALPUFF takes three-dimensionally varying wind, temperature and turbulence fields from the CALMET model. The 3-D winds and temperature fields from CALMET are reconstructed using meteorological measurements, orography and land use data. Besides wind and temperature fields, CALMET determines the 2-D fields of micrometeorological variables needed to carry out dispersion simulations (mixing height, Monin Obukhov length, friction velocity, convective velocity, etc.). A two-step approach is typically used to compute the wind fields in CALMET. In the first step, an initial guess wind field is adjusted for kinematic effects of terrain, slope flows, and terrain blocking effects to produce a Step 1 wind field. The second step applies an objective analysis procedure to introduce observational data into the Step 1 wind fields to produce the final wind fields. In this study, CALMET used the Weather Research and Forecasting (WRF) model due to its capability of simulating regional flows and certain aspects of local meteorological conditions such as complex terrain. It replaces the two-step approach because of the higher spatial resolution of the WRF output compared to observational data. The output of the CALMET model is directly interfaced with the CALPUFF dispersion model for further air quality modelling.

A paragraph summarizing the PAC emissions will be included in the beginning of the Results and Discussion (sect. 3.1) and is provided below:

3.1 PAC emissions estimates

Unsubstituted and alkylated PAHs and DBTs emissions from oil sands development and non-industrial sources were estimated over the model domain (Table 1). Total unsubstituted PAH emissions (2009-2014) are estimated to be 56 to 58 tonnes yr-1 based on emissions from tailings ponds, mine face, mine fleet, residential, commercial, local traffic, airport, point, and transportation sources. Point sources accounted for most of the total unsubstituted PAH emissions (75-77%). The major difference in the total unsubstituted PAH emissions between the CEMA-derived and JOSM-derived emissions databases is the higher evaporative PAH emissions from tailings ponds and mine face in the JOSM-derived emissions database. Alkylated PAH and DBT emissions (2011-2014) are estimated to be 17 tonnes yr-1 and 0.26 tonnes yr-1 respectively; however, they consisted of fewer emission sources (tailings ponds, mine fleet and transportation sources) due to a lack of PAC speciation data to estimate emissions from other sources. Nevertheless, the PAC emissions estimates in this study may still be underestimated, e.g. tailings pond and fugitive dust emissions. A common technique for measuring tailings pond emissions is a flux chamber. However, recent studies suggest that this technique underestimates organic compound emission fluxes (Tran et al., 2018). Windblown petcoke dust observed recently over surface mining areas in the AOSR (Zhang et al., 2016) also have not been accounted for in the PAC emissions databases.

Additionally, PAHs have a wide spectrum including compounds in gaseous phase and particulate phase, which can exhibit different characteristics during transport and deposition. Although the research is focused on relative comparison of two emissions databases with only considering dispersion, the author may analyze qualitatively the resultant impact of turning off deposition modelling on modelling results in general.

Response: We decided to present the model results without deposition processes after running a few model scenarios. One of the impacts of turning off deposition modeling is that the modeled air concentrations are higher compared to those with deposition modeling turned on. However, we found that the modeled concentrations from simulating emissions, transport and dispersion processes, but without deposition processes, are already lower than measurements, demonstrating that the emissions inputs are conservative or underestimated. If deposition processes were included, modeled concentrations would be even lower than measurements; however, in this model scenario it would be hard to say if this was caused by too low emissions input or too high deposition rates, knowing that large uncertainties exist in treating dry and wet deposition processes. For example, there are large uncertainties in the PAC dry deposition velocities (Zhang et al., 2015a), PAC scavenging ratios for snow and rain scavenging of gas-phase and particulate-phase PACs (Zhang et al., 2015b), and scavenging coefficients of aerosols in general by snow and rain scavenging processes (Zhang et al., 2013; Wang et al., 2014). Our next study related to this project is to compare the deposition output using various approaches.

References:

Tran, H. N. Q., Lyman, S. N., Mansfield, M. L., O'Neil, T., Bowers, R. L., Smith, A. P. and Keslar, C.: Emissions of Organic Compounds from Produced Water Ponds II: Evaluation of flux-chamber measurements with inverse-modeling techniques, J. Air Waste Manag. Assoc., DOI: 10.1080/10962247.2018.1426654, 2018.

Wang, X., Zhang, L., and Moran, M. D.: Development of a new semi-empirical parameterization for below-cloud scavenging of size-resolved aerosol particles by both rain and snow, Geosci. Model Dev., 7, 799-819, 2014.

Zhang, L., Cheng, I., Wu, Z., Harner, T., Schuster, J., Charland, J. P., Muir, D., and Parnis, J.M.: Dry deposition of PACs to various land covers in the Athabasca Oil Sands Region, J. Adv. Model. Earth Sy., 7, 1339-1350, 2015a.

Zhang, L., Cheng, I., Muir, D., and Charland, J.-P.: Scavenging ratios of polycyclic aromatic compounds in rain and snow in the Athabasca oil sands region, Atmos. Chem. Phys., 15, 1421-1434, 2015b.

Zhang, L., Wang, X., Moran, M. D., and Feng, J.: Review and uncertainty assessment of size-resolved scavenging coefficient formulations for below-cloud snow scavenging of atmospheric aerosols, Atmos. Chem. Phys., 13, 10005-10025, 2013.

Zhang, Y., Shotyk, W., Zaccone, C., Noernberg, T., Pelletier, R., Bicalho, B., Froese, D. G., Davies, L. and Martin, J. W., 2016: Airborne petcoke dust is a major source of polycyclic aromatic hydrocarbons in the Athabasca Oil Sands Region, Environ. Sci. Technol., 50(4), 1711-1720, 2016.

---

## Author Response (AR1)

Dear Dr. Brook (Co-Editor):

We are submitting a revised paper (manuscript #acp-2017-1091) entitled, *Emissions databases for polycyclic aromatic compounds in the Canadian Athabasca Oil Sands Region – development using current knowledge and evaluation with passive sampling and air dispersion modelling data*, for further consideration in Atmospheric Chemistry and Physics. We have addressed all of the comments provided by the three reviewers. The details can be found in the enclosed response to reviewers' comments. For your convenience, a copy of the paper with track changes is also enclosed.

Thank you for taking care of the review process for this paper.

Sincerely,

Xin Qiu, Irene Cheng, Fuquan Yang, Erin Horb, Leiming Zhang, and Tom Harner

**Response to Reviewers' Comments**

*Emissions databases for polycyclic aromatic compounds in the Canadian Athabasca Oil Sands Region – development using current knowledge and evaluation with passive sampling and air dispersion modelling data (acp-2017-1091)*

**Referee #1**

We appreciate the comments by the reviewer to help us improve the paper. Our responses to the specific comments are shown below in blue.

This paper developed two speciated and spatially-resolved emissions databases for polycyclic aromatic compounds (PAC) in the Athabasca oil sands region (AOSR), and compared the two emissions databases with the measurements from a passive air monitoring network. Papers have a high degree of novelty and I recommend to publish after a minor revised.

1. Please delete some basic concepts, concise articles.

Response: We condensed and deleted some of the basic information (e.g. basics about PAHs in the first paragraph of the introduction) in the revised paper to keep it concise.

2. What are the PAHs in CEMA database and JOSM database, respectively? Are they the same?

Response: The 16 parent PAHs are the same in the CEMA-derived and JOSM-derived emissions databases. As mentioned in sect. 2.2, the PAHs include: naphthalene (NAPH), acenaphthylene (ACY), acenaphthene (ACE), fluorene (FLR), phenanthrene (PHEN), anthracene (ANTH), fluoranthene (FLRT), pyrene (PYR), benz[a]anthracene (BaA), coeluting chrysene and triphenylene (CHRYþTRIP), benzo[b]fluoranthene (BbF), benzo[k]fluoranthene (BkF), benzo[a]pyrene (BaP), indeno[1,2,3-cd]pyrene (I123cdP), dibenz[a,h]anthracene (dBahA) and benzo[ghi]perylene (BghiP). However, total alkylated PAHs and dibenzothiophenes (DBT) concentrations were modelled for the JOSM-derived emissions scenario only, since the monitoring of these additional compounds is part of monitoring activities under the JOSM program. This is clarified in sect. 2.2 (2nd paragraph) of the revised paper.

3. In Results and Discussion, please describe the same meteorological data in detail.

Response: A more detailed description of the meteorological data input for the CALMET model is provided in sect. 2.2 (1st paragraph) of the revised paper.

CALPUFF takes three-dimensionally varying wind, temperature and turbulence fields from the CALMET model. The 3-D winds and temperature fields from CALMET are reconstructed using meteorological measurements, orography and land use data. Besides wind and temperature fields, CALMET determines the 2-D fields of micrometeorological variables needed to carry out dispersion simulations (mixing height, Monin Obukhov length, friction velocity, convective velocity, etc.). A two-step approach is typically used to compute the wind fields in CALMET.

In the first step, an initial guess wind field is adjusted for kinematic effects of terrain, slope flows, and terrain blocking effects. The second step applies an objective analysis procedure to introduce observational data into the first step to produce the final wind fields. In this study, CALMET used the Weather Research and Forecasting (WRF) model due to its capability of simulating regional flows and certain aspects of local meteorological conditions such as complex terrain. It replaces the two-step approach given the higher spatial resolution of the WRF output compared to observational data. The output of the CALMET model is directly interfaced with the CALPUFF dispersion model for further air quality modelling.

**Referee #2**

We appreciate the comments by the reviewer to help us improve the paper. Our responses to the specific comments are shown below in blue.

Within the article, two speciated and spatially-resolved emissions databases for PACs in AOSR were developed. Further, the PAC concentrations in AOSR were simulated using the CALPUFF atmospheric dispersion model for both scenarios (both databases) and compared with passive monitoring data to assess which emissions input can achieve better agreement with measurements. According to my opinion, the manuscript represent a significant scientific contribution in studying PACs (PAHs, alkylated PAHS and DBTs) in oil sends regions where uncertainties in the PACs emissions are still significant. I recommend the manuscript for publication with minor revision:

1. Although if deposition had been considered in the CALPUFF model, the modeled values would be even lower then the measured, I would ask the authors to explain why they have excluded the loss by wet and dry deposition in the modeling process. Were there any other reasons?

Response: We made this decision after running a few model scenarios and decided to present the model results without deposition processes. This is because modeled concentrations from simulating emissions, transport and dispersion processes, but without deposition processes, are already lower than measurements, demonstrating that emission inputs are conservative or underestimated. By including deposition processes, modeled concentrations would be even lower than measurements; however, in this model scenario it would be hard to say if this was caused by too low emissions input or too high deposition rates, knowing that large uncertainties exist in treating dry and wet deposition processes. For example, there are large uncertainties in the PAC dry deposition velocities (Zhang et al., 2015a), PAC scavenging ratios for snow and rain scavenging of gas-phase and particulate-phase PACs (Zhang et al., 2015b), and scavenging coefficients of aerosols in general by snow and rain scavenging processes (Zhang et al., 2013; Wang et al., 2014). Our rationale for excluding deposition loss in the model is discussed in the

paragraph before sect. 3.3 of the revised paper. In our next study related to this project, we plan on comparing the deposition output using various approaches.

2. The authors should also take into consideration the fact that values of PAC concentrations obtained using the passive samplers refer only to the gaseous phase of pollutant and reflect a more accurate concentration for the low molecular weight PACs compering to high molecular weight compounds.

Response: Although certain types of passive air samplers show a bias for gas-phase compounds, the PUF disk samplers used in the oil sands network have been shown to capture both gas-phase and particle-phase PAHs with the same efficiency as conventional high volume air samplers (Harner et al., 2013; Markovic et al., 2015).

**References:**

Harner, T., Su, K., Genualdi, S., Karpowicz, J., Ahrens, L., Mihele, C., Schuster, J., Charland, J. -P. and Narayan, J.: Calibration and application of PUF disk passive air samplers for tracking polycyclic aromatic compounds (PACs), Atmos. Environ., 75, 123-128, 2013.

Markovic, M., Prokop, S., Staebler, R.M., Liggio, J., Harner, T: Evaluation of the particle infiltration efficiency of three passive samplers and the PS-1 active air sampler, Atmos. Environ., 112, 289-293, 2015.

Wang, X., Zhang, L., and Moran, M. D.: Development of a new semi-empirical parameterization for below-cloud scavenging of size-resolved aerosol particles by both rain and snow, Geosci. Model Dev., 7, 799-819, 2014.

Zhang, L., Cheng, I., Wu, Z., Harner, T., Schuster, J., Charland, J. P., Muir, D., and Parnis, J.M.: Dry deposition of PACs to various land covers in the Athabasca Oil Sands Region, J. Adv. Model. Earth Sy., 7, 1339-1350, 2015a.

Zhang, L., Cheng, I., Muir, D., and Charland, J.-P.: Scavenging ratios of polycyclic aromatic compounds in rain and snow in the Athabasca oil sands region, Atmos. Chem. Phys., 15, 1421-1434, 2015b.

Zhang, L., Wang, X., Moran, M. D., and Feng, J.: Review and uncertainty assessment of sizeresolved scavenging coefficient formulations for below-cloud snow scavenging of atmospheric aerosols, Atmos. Chem. Phys., 13, 10005-10025, 2013.

**Referee #3**

We appreciate the comments by the reviewer to help us improve the paper. Our responses to the specific comments are shown below in blue.

This research work compared CALPUFF modelling results applying the two air emissions databases of CEMA and JOSM programs. The modelling results are then compared with observations to evaluate accuracy of the air emissions values. This research makes significant contribution to the work of PAHs air emission estimation in the oil sands region. While dispersion models could have systematic error existing inherently in the model, particularly and usually lead to underestimation at low pollutant concentrations, this research presents a progressive approach to compare the modelling results relatively for the original emissions data and the improved one. I suggest to publish it to make colleagues working in this field be aware of the work progress.

It would be clearer if the author could add more information on meteorology and emission summary.

Response: A summary of the meteorological model that drives the CALPUFF model is provided in sect. 2.2 (1st paragraph) of the revised paper. CALPUFF takes three-dimensionally varying wind, temperature and turbulence fields from the CALMET model. The 3-D winds and temperature fields from CALMET are reconstructed using meteorological measurements, orography and land use data. Besides wind and temperature fields, CALMET determines the 2-D fields of micrometeorological variables needed to carry out dispersion simulations (mixing height, Monin Obukhov length, friction velocity, convective velocity, etc.). A two-step approach is typically used to compute the wind fields in CALMET. In the first step, an initial guess wind field is adjusted for kinematic effects of terrain, slope flows, and terrain blocking effects. The second step applies an objective analysis procedure to introduce observational data into the first step to produce the final wind fields. In this study, CALMET used the Weather Research and Forecasting (WRF) model due to its capability of simulating regional flows and certain aspects of local meteorological conditions such as complex terrain. It replaces the two-step approach given the higher spatial resolution of the WRF output compared to observational data. The output of the CALMET model is directly interfaced with the CALPUFF dispersion model for further air quality modelling.

A paragraph summarizing the PAC emissions is included in sect 3.1 of the revised paper and also provided below. We also added a map showing the spatial distribution of the unsubstituted PAH emissions in Fig. S3 of the Supplement.

**3.1 PAC emissions estimates**

Over the model domain, the total unsubstituted PAH emissions (2009-2014) are estimated to be 56 to 58 tonnes  $yr^{-1}$  based on emissions from tailings ponds, mine face, mine fleet, residential,

commercial, local traffic, airport, point, and transportation sources (Table 1). A map of the spatial distribution of the emissions is shown in Fig. S3 of the Supplement. Point sources accounted for most of the total unsubstituted PAH emissions (75-77%). The major difference in the total unsubstituted PAH emissions between the CEMA-derived and JOSM-derived emissions databases is the higher evaporative PAH emissions from tailings ponds and mine face in the JOSM-derived emissions database. Alkylated PAH and DBT emissions (2011-2014) are estimated to be 17 tonnes yr-1 and 0.26 tonnes yr-1 respectively; however, they consisted of fewer emission sources (tailings ponds, mine fleet and transportation sources) due to a lack of PAC speciation data to estimate these emissions. Nevertheless, the PAC emissions estimates may still be underestimated from oil sands sources, such as tailings ponds and fugitive dust. Recent studies suggest that flux chamber measurements of tailings pond emissions underestimate organic compound emission fluxes (Tran et al., 2018). Windblown petcoke dust observed recently over surface mining areas in the AOSR (Zhang et al., 2016) also have not been accounted for in the PAC emissions databases. In addition to gaps in the existing emissions databases, speciation profiles were largely missing particularly for alkylated PAHs and DBTs. In this paper, the focus is on unsubstituted PAHs; however, alkylated PAHs and DBTs were still modelled despite the limited knowledge of the emissions profiles.

Additionally, PAHs have a wide spectrum including compounds in gaseous phase and particulate phase, which can exhibit different characteristics during transport and deposition. Although the research is focused on relative comparison of two emissions databases with only considering dispersion, the author may analyze qualitatively the resultant impact of turning off deposition modelling on modelling results in general.

Response: We decided to present the model results without deposition processes after running a few model scenarios. One of the impacts of turning off deposition modelling is that the modelled air concentrations are higher compared to those with deposition modelling turned on. However, we found that the modeled concentrations from simulating emission, transport and dispersion processes, but without deposition processes, are already lower than measurements, demonstrating that the emissions inputs are conservative or underestimated. If deposition processes were included, modeled concentrations would be even lower than measurements; however, in this model scenario it would be hard to say if this was caused by too low emissions input or too high deposition rates, knowing that large uncertainties exist in treating dry and wet deposition processes. For example, there are large uncertainties in the PAC dry deposition velocities (Zhang et al., 2015a), PAC scavenging ratios for snow and rain scavenging of gas-phase and particulate-phase PACs (Zhang et al., 2015b), and scavenging coefficients of aerosols in general by snow and rain scavenging processes (Zhang et al., 2013; Wang et al., 2014). This discussion is provided in the paragraph before sect. 3.3 of the revised paper. In our next study related to this project, we plan on comparing the deposition output using various approaches.

1Novus Environmental Inc., Guelph, Ontario, N1G 4T2, Canada

[revised manuscript text omitted]

na: not available

Figure 1: Average total PAH concentrations (excluding NAPH) from November 2010 to June 2012 at local (a) and remote sites (b): passive measurements (blue), modelled-JOSM case (striped) and modelled-CEMA case (grey).